# Suicide Risk in Post-COVID-19 Syndrome

**DOI:** 10.3390/jpm12122019

**Published:** 2022-12-07

**Authors:** Alessio Simonetti, Evelina Bernardi, Delfina Janiri, Marianna Mazza, Silvia Montanari, Antonello Catinari, Beatrice Terenzi, Matteo Tosato, Vincenzo Galluzzo, Francesca Ciciarello, Francesco Landi, Gabriele Sani

**Affiliations:** 1Menninger Department of Psychiatry and Behavioral Sciences, Baylor College of Medicine, Houston, TX 77030, USA; 2Department of Neuroscience, Section of Psychiatry, Fondazione Policlinico Universitario Agostino Gemelli IRCCS, 00168 Rome, Italy; 3Department of Neurology and Psychiatry, Sapienza University of Rome, 00185 Rome, Italy; 4Department of Geriatrics, Università Cattolica del Sacro Cuore, 00168 Rome, Italy; 5Department of Geriatrics, Fondazione Policlinico Universitario A. Gemelli IRCCS, 00168 Rome, Italy; 6Department of Neurosciences, Section of Psychiatry, Università Cattolica del Sacro Cuore, 00168 Rome, Italy

**Keywords:** suicide, depression, psychopathology, COVID-19, post-COVID-19 syndrome

## Abstract

Post-acute sequelae of COVID-19 include several neuropsychiatric disorders. Little is known about the relationship between post-COVID-19 syndrome and suicidality. The aim of the study was to investigate the risk of suicide in subjects with persistent post-COVID-19 syndrome. One-thousand five-hundred eighty-eight subjects were assessed in the Post-Acute Care Service at the Fondazione Policlinico Universitario “Agostino Gemelli” IRCCS of Rome. Assessment included: (a) sociodemographic characteristics; (b) symptoms during and after COVID-19; (c) psychopathological evaluation. Participants were divided in those with (SUI) or without (NON SUI) suicide risk according to the Mini International Neuropsychiatric Interview. Additionally, subjects with SUI were split into those with high (HIGH SUI) and low (LOW SUI) suicide risk. Between-group comparisons were made with *t*-tests for continuous variables and χ^2^ tests for categorical variables. SUI showed greater percentages of physical complaints during and after COVID-19, greater percentages of psychiatric history and presence of psychiatric history in relatives, greater percentages of subjects previously undergoing psychopharmacotherapy, and greater levels of anxiety, mixed depressive symptoms, general psychopathology than NON SUI. HIGH SUI showed greater number of symptoms during and after COVID-19 and higher levels of mixed depressive symptoms than LOW SUI. Percentages of subjects undergoing psychotherapy was higher in LOW SUI than HIGH SUI. Greater levels of physical complaints and psychopathology during post-COVID-syndrome might enhance the risk of committing suicide. Treatment of physical complaints and psychotherapy might reduce suicide risk.

## 1. Introduction

Coronavirus-19 disease (COVID-19) is a respiratory syndrome caused by severe acute respiratory syndrome-Coronavirus-2 (SARS-CoV-2). The first cases of COVID-19 were reported at the end of 2019 in Wuhan, China [1]. Since then, infection from SARS-CoV-2 has spread globally, officially becoming a pandemic on 11 March 2020 [2]. COVID-19 symptoms show a higher clinical variability: from asymptomaticity to severely symptomatic forms [3]. Severe SARS-CoV-2 infection might cause death due to pneumonia, acute respiratory distress syndrome (ARDS), and multiple organ failure. Physical distress brought by COVID-19 does not generally extinguish with the end of the infection. A substantial share of people report clinical sequelae in the weeks or months following symptom onset. These symptoms, known as post-COVID syndrome, post-COVID-19 syndrome, or long-COVID syndrome, include cough, dyspnea, fatigue, difficulties with memory and concentration (“brain fog”), sleep disorders, gastrointestinal complaints, and musculoskeletal problems [4,5]. Additionally, psychiatric syndromes, such as depression, anxiety, post-traumatic disorders, and sleep disturbances, have been frequently reported [6]. Among the psychiatric symptoms related to the COVID-19 pandemic, suicide represents one of the major public health concerns worldwide [7]. Suicide rates have been related to depression and related psychopathology, such as anxiety [8], presence of mixed depressive states [9], and hopelessness [10]. After the height of the COVID-19 pandemic, the attention given to suicidal behaviors has steeply risen because of the high rates of suicide observed in other coronavirus-mediated ARDSs [11,12,13]. As a consequence, several studies have investigated suicide ideation/attempt rates during the COVID-19 pandemic, using online surveys, electronic records, and health system registries. Five studies [14,15,16,17,18,19] focused on possible risk factors for suicidality. Social distancing due to quarantine, unemployment, safety concerns, being unmarried, and previous psychopathology were found to be the most associated with suicide.

Others [20,21,22,23,24] mainly focused on recommendations for suicide prevention. Strategies proposed were: (I) developing a systematic suicide screening test; (II) facilitating communication and increasing access to interventions for the people at risk; (III) promoting mental health programs; (IV) implementing measures to mitigate the economic breakdown; (V) regulating media reporting. Evidence on the increased/decreased risk of suicide during the COVID-19 pandemic is still conflicting [7,25]. Nevertheless, the literature focusing on the specific relationship between post-COVID-19 syndrome and suicide risk is lacking, and to date, only one study has investigated this relationship [26].

In the study by Gasnier and colleagues, cognitive complaints were associated with greater suicide risk. Reliability of the aforementioned study is hampered by the relatively small sample size. Additionally, the number of post-COVID-19 symptoms taken into account were limited to fatigue, respiratory and cognitive complaints, headache, paresthesia, anosmia, weakness, and pain. Therefore, larger samples are needed to better clarify the relationship between post-COVID-19 syndrome and suicide risk. Additionally, a greater number of post-COVID-19 syndrome symptoms should be included while investigating this relationship. The present study’s aim is to clarify these aspects through the comparison of a large sample of subjects with post-COVID-19 syndrome, with or without suicide risk. According to the aforementioned relationship between psychopathology, SARS-related symptoms, and suicide risk, and the previous work of Gasnier and colleagues, we expected that greater levels of psychopathology, a greater number of physical symptoms related to post-COVID-19 syndrome, and greater levels of cognitive complaints were present in subjects with suicide risk than those without.

## 2. Materials and Methods

### 2.1. Sample

The study was conducted by all the authors in the context of the Gemelli Against COVID-19 Post-Acute Care Study Group. The group was formed with the aim of studying the post-acute sequelae of SARS-CoV2 infection. The sample consisted of a cohort of subjects who had contracted SARS-CoV-2 infection and underwent a multidisciplinary evaluation in the Post-Acute Care Service at the Fondazione Policlinico Universitario “Agostino Gemelli” IRCCS of Rome, Italy (Gemelli Against COVID-19 Post-Acute Care Service), from 21 April 2020 to 11 May 2022. Patients underwent a comprehensive assessment, including collection of detailed medical history and a thorough physical examination. Additionally, they received internal medicine, geriatric, ophthalmological, otolaryngologic, pneumological, psychiatric, cardiological, immunological, and rheumatological evaluations. Specifically, anamnesis regarding demographical characteristic as well as severity and course of COVID-19 infection, together with a general medical examination, was performed by an internal medicine specialist. Geriatric, ophthalmological, otolaryngologic, pneumological, psychiatric, cardiological, immunological, and rheumatological evaluations were provided by specialists of the Gemelli Against COVID-19 Post-Acute Care Service. As regards psychiatric evaluation, a senior psychiatrist (AS) performed diagnostic assessments using rating scales assessing depression, suicidal risk, severity of general psychopathology, severity of depressive symptoms, severity of anxiety symptoms, severity of mixed depression, and hopelessness. Then, a minimum of two trained diagnosticians (postdoctoral researchers) were additionally involved in the diagnostic process. Specifically, diagnostic consensus among the treating psychiatrist, the senior psychiatrist, and the trained diagnosticians was needed to include the subject in the study. In the event of diagnostic disagreement between one or more subjects belonging to the diagnostic team, the participant was excluded from the study. Prior to the start of the study, specific training on rating scale administration was provided. Then, psychometricians performed practice assessments and received performance reviews and feedback on test administration. Their training process continued until an inter-rater reliability of at least 0.80 (κ coefficient) was reached.

Inclusion criteria were (a) age between 18 and 75; (b) previous positivity to COVID-19; (c) capability of providing informed consent. Exclusion criteria were severe neurodevelopmental disorders, dementia, or other severe neurological disorders.

Two-thousand five-hundred fifty-nine subjects accessed the Post-Acute Care Service of the Fondazione Policlinico Universitario Agostino Gemelli IRCCS. Two-hundred nine patients were excluded due to unsuitable age range. Six subjects were excluded because of dementia. Seven-hundred fifty-six subjects were unable to complete the whole evaluation and were excluded from the study. Therefore, the final sample consisted of 1588 subjects.

In accordance with published research [27,28,29,30,31,32] and due to the lack of homogeneity in the proposed criteria for the definition of the post-COVID-19 syndrome [33,34], we defined as Post-COVID-19 syndrome any condition characterized by the presence of one or more symptoms that persisted after negativization, regardless of symptom duration or severity. Subjects involved gave their written informed consent to the study, and the study was approved by the Ethical Committee of the Fondazione Policlinico Universitario Agostino Gemelli IRCCS (protocol number: 0013008/20).

### 2.2. Assessment

Data collected for the present study include (a) sociodemographic characteristics (i.e., age, gender, presence/absence of employment, education); (b) data regarding SARS-CoV-2 infection, i.e., time elapsed from the infection, symptoms during the infection, presence/absence of hospitalization, hospitalization length, presence/absence of pharmacotherapy; (c) data regarding comprehensive assessment made in the Post-Acute Care Service at the Fondazione Policlinico Universitario “Agostino Gemelli” IRCCS of Rome, i.e., data regarding post-COVID-19 syndrome; (d) data regarding psychiatric evaluation, i.e., presence/absence of psychiatric history, presence/absence of psychiatric history in the first-degree relatives, number of psychotropic drugs assumed, and presence of psychopathology as assessed with rating scales. Specifically, the following domains were investigated: suicidal risk, severity of general psychopathology, severity of depressive symptoms, severity of anxiety symptoms, severity of mixed depression, and hopelessness. These domains were assessed with rating scales largely used in common clinical practice as well as in research. The aforementioned psychopathologic assessment was chosen because of the close relationship to suicide risk and attempts [8,9,10]. The detailed description of psychiatric rating scales used is provided below.

The suicidal subscale of the Mini International Neuropsychiatric Interview (MINI-suicidal) [35]: MINI-suicidal is a subscale of the Mini International Neuropsychiatric Interview (MINI) [36], i.e., a short, structured clinic interview that allows diagnosis of psychiatric disorders according to the Diagnostic and Statistical Manual of Mental Disorders (DSM) [37] and the International Classification of Mental and Behavioral Disorders (ICD-10) [38]. MINI-suicidal assesses suicidal risk. We used the short form, which was found to be more suitable for screening in an acute setting. It consists of six items that are scored yes or no. Items 1–5 refer to suicidal thoughts/behaviors occurring during the last month, and item 6 records suicide lifetime occurrence. Each item score is weighted according to its estimated contribution to suicide risk. The total score range is 0–33.

The Brief Psychiatric Rating Scale (BPRS) [39]: BPRS was developed as a measurement of general psychopathology. It assesses a large variety of psychiatric symptoms, such as anxiety, depression, and psychosis. The presence and severity of psychiatric symptoms were rated on a Likert scale ranging from 1 (not present) to 7 (extremely severe). Possible scores vary from 24 to 168, with lower scores indicating less severe psychopathology.

The Hamilton Rating Scale for Depression (HAM-D) [40]: HAM-D is used to assess the severity of depressive symptoms. We used the original version, containing 17 items. Each item is related to depressive symptoms experienced over the past week. Each item is scored between 0–4 or 0–2, depending on the item. Scores of 0–7 are indicative of the absence of depression, scores of 8–16 suggest mild depression, scores of 17–23 are indicative of moderate depression, and scores over 24 are indicative of severe depression.

The Hamilton Anxiety Rating Scale (HAM-A) [41]: HAM-A is a questionnaire used to measure the severity of anxiety symptoms. The scale consists of 14 items and measures both psychic anxiety (worries, irritability, feelings of tension, etc.) and somatic anxiety (physical complaints related to anxiety). Total scores range between 0 and 56. Scores < 17 indicate mild anxiety, 18–24 mild to moderate anxiety, 25–30 moderate to severe anxiety, and >30 severe anxiety.

The Koukopoulos Mixed Depression Rating Scale (KMDRS) [42]: KMDRS is a self-administered rating scale, consisting of 14 items evaluating the presence and severity of the typical symptoms of mixed depression. Possible scores range from a minimum of 0 to a maximum of 51, with higher scores indicating a greater severity of mixed depressive symptoms.

The Beck Hopelessness Scale (BHS) [43]: BHS is a self-assessment questionnaire that consists of 20 true–false items concerning three factors: feelings about the future, loss of motivation, and future expectations. The total BHS score can range from 0 to 20, such that higher scores reflect higher levels of hopelessness. Total scores of 0–3 are in the normal range, 4–8 suggest mild hopelessness, 9–14 suggest moderate hopelessness, and scores greater than 14 identify severe hopelessness. BHS is composed of three subscales: BHS-future, assessing feelings about the future; BHS-motivation, assessing loss of motivation; BHS expectation, assessing future expectations.

The sample was divided into two groups according to MINI-suicide cutoffs: (1) subjects without suicide risk (NON SUI; total MINI-suicide score = 0); (2) subjects with suicide risk (SUI; total MINI-suicide score ≥ 1). We chose this cutoff in accordance with the vast majority of studies on suicide, which identify total scores ≥ 1 as indicative of the presence of suicide risk [44,45].

In case of any suicide risk, we provided tailored care in accordance with the European Psychiatric Association Guidelines for Suicide Prevention [46]. Specifically, we immediately evaluated the patient and implemented a treatment plan focused on reducing acute psychiatric symptoms such as anxiety, insomnia, depression, and eventual psychotic symptoms. In severe cases or when the patient was not surrounded by a social network, we opted for hospitalization in a psychiatric ward.

When we could send the patient home, we involved family members in the treatment plan, asking them to not leave the patient alone. Then, we increased surveillance by providing more frequent visits and telephone contacts to monitor progress and the possible side effects of medication. Adjustment of the aforementioned guidelines was made according to COVID-19 restrictions [47].

### 2.3. Statistical Analysis

Descriptive analyses of the sample were initially investigated.

Between-group differences were analyzed with the chi-squared test (χ^2^) for nominal variables and *t*-tests for continuous variables.

In each t/χ^2^-test, groups (NON SUI; SUI) were independent variables, while sociodemographic characteristics (age, gender, presence/absence of employment, education), data regarding SARS-CoV-2 infection and post-COVID-19 syndrome (time elapsed from COVID-19, type and number of symptoms present during COVID-19, presence/absence of hospitalization due to COVID-19, hospitalization length, number and type of symptoms during the evaluation; and during post-COVID-19 syndrome, number and types of current medications), data regarding psychiatric evaluation (presence/absence of psychiatric history, presence/absence of psychiatric history in the first-degree relatives, number of past psychotropic drugs, presence/absence of previous psychotherapy, number of psychotropic drugs at the time of evaluation, psychopathological scales total scores) were dependent variables.

Additional exploratory analyses were performed to investigate suicide risk: SUI were further divided into subjects at low risk (LOW SUI; total MINI score < 5) and subjects at moderate, high risk (HIGH SUI, MINI score ≥ 6), in accordance with the study on the predictive validity of the MINI suicidal subscale [36]. T/χ^2^-tests were performed to test between-group differences. Variables taken into account were limited to who differed between SUI and NON SUI. Cohen’s d and Cramer’s V were calculated to investigate effect sized for continuous and discrete variables, respectively.

We used the statistical routines of SPSS Statistics 24.0 for Windows (IBMCo., Armonk, NY, USA, 2016).

## 3. Results

Members of the sample had a mean age of 55.27 ± 14.35 years; they were mostly men (*n* = 839, 52,8%), most (*n* = 1049, 66.1%) had an occupation, and they had a mean of 13.36 ± 5.41 years of education.

SUI characteristics are summarized in Table 1. Within the sample (*n* = 1588), 41 subjects (2.6%) had a suicidal risk. Nineteen subjects (46.3%) had a low risk, and 22 (53.6%) had a high risk. In SUI, 36 subjects (87.8%) had a diagnosis of a major depressive disorder (MDD); 13 subjects (31.7%) had a psychiatric history. Twenty-one subjects (51.2%) recently thought about suicide, 12 (29.3%) declared to have a specific suicidal plan, and 3 (7.3) attempted suicide.

As regards sociodemographic characteristics, SUI showed higher rates of women than NON SUI. No other difference emerged (see Table 2).

As regards differences in data related to SARS-CoV-2 infection and post-COVID syndrome, SUI showed greater frequency of COVID-19–associated symptoms. Such differences are mainly driven by higher rates of diarrhea, vertigo, joint pain, Raynaud’s phenomenon, myalgia, and chest pain. In addition, SUI showed greater number of symptoms at the time of evaluation (i.e., symptoms related to post-COVID-19 syndrome). Such difference was mainly due to the higher rates of fever, fatigue, cough, diarrhea, headache, anosmia, vertigo, joint pain, skin lesions, Sjogren syndrome, and myalgia. SUI showed higher percentages of subjects receiving medications. Such differences were driven by the higher percentages of subjects receiving endocrinological and pneumological medications. Between-group differences are shown in Table 3.

Differences regarding psychopathology are present in Table 4. SUI showed higher rates of psychiatric history, psychiatric history in first-degree relatives, and previous consumption of psychotropic drugs than NON SUI, whereas differences in rates of subjects under treatment with psychotropic drugs only approached difference. SUI more likely underwent psychotherapy and showed higher BPRS, HAM-D, HAM-A, and KMDRS total scores than NON SUI. SUI also showed higher scores of BHS-future and BHS-motivation than NON SUI.

Effect sizes regarding differences in psychopathology were strong, whereas those regarding number of symptoms during and after COVID-19 were moderate. On the other hand, effect sizes related to differences in demographical and clinical variables and specific COVID-19 and post-COVID-19 syndrome symptoms were small.

Differences regarding HIGH SUI and LOW SUI are shown in Table 2, Table 3 and Table 4. HIGH SUI showed a higher number of symptoms at the time of evaluation. Such difference was mainly driven by higher rates of subjects suffering from fatigue, Sjogren’s syndrome, and myalgia in HIGH SUI than LOW SUI. HIGH SUI showed lower rates of subjects with psychiatric history in first-degree relatives and lower percentages of subjects undergoing psychotherapy than LOW SUI. HIGH SUI showed higher total scores of BPRS and KMDRS than LOW SUI.

## 4. Discussion

Results can be summarized as follows: (a) Subjects with post-COVID-19 syndrome and suicide risk showed a greater number of physical symptoms related to COVID-19 and post-COVID-19 syndrome; (b) Subjects with post-COVID-19 syndrome and suicide risk showed higher rates of psychopathology than those without suicide risk; (c) the number of symptoms associated with post-COVID-19 also differentiated subjects at high risk from those at low risk. Differences in psychopathology only involve severity of general psychopathology and severity of mixed depression. Furthermore, a shorter percentage of subjects with high risk of suicide underwent psychotherapy as compared with those having a low risk.

The present sample’s rates of suicide risk and rates of suicidal thoughts and attempts were not increased as compared to the general population [48]. They are also discordant with recent studies reporting an increase of suicidal thoughts and deaths during COVID-19 [25,49]. The patients’ enrollment’s timeframe might account for discrepancies found. The larger part of patients’ enrollment ranged from April 2020 to April 2021, i.e., the first year of COVID-19 pandemic. Specifically, 1261 subjects (81.5% of the whole sample) were assessed in this period.

Accordingly, other studies reported similar [50] or reduced [21,51,52,53,54] rates of suicide in the early phase of COVID-19 as compared with periods before the pandemic. Such findings might be due to the “honeymoon period” and “pulling together” phenomena seen in prior natural disasters [55,56]. Following these models, in the first periods of a natural disaster, individuals display a positive affect, focusing less on themselves and more on the disaster response [54]. As a result, the periods immediately following the disaster might be related to a general relief and therefore to lower rates of suicide.

The higher rates of women we found within SUI than NON SUI might seem in contrast with the well-known finding of a higher suicide rate in men as compared to women [57]. Such a discrepancy has been described before as the “gender paradox in suicide”. The paradox resides in the fact that females show more suicidal thoughts and attempt suicide more frequently than males do, whereas males have higher rates of suicide [58]. As shown by a Japanese study of Takubo and colleagues on suicidal ideation among postpartum mothers during COVID-19 [59], this phenomenon has been exacerbated by the pandemic, maybe due to the decrease in female employment and increase in domestic violence [53].

Differently from what was expected and from the findings of Gasnier and colleagues, we did not find any effect of cognitive complaints on suicide risk. Possible reasons for such a discrepancy might involve the type of cognitive assessment tools used. Gasnier and colleagues used the Montreal Cognitive Assessment Scale, i.e., a structured battery investigating a large array of cognitive symptoms such as short-term memory, visuospatial abilities, executive functions, attention, concentration, working memory, language, and orientation to time and place [60]. Neurocognitive evaluation of the present study concerns only subjective complaints of cognitive difficulties and is also limited to the domains of attention and concentration. Therefore, a more comprehensive evaluation of cognitive status is needed to investigate the predictive effect of cognitive alterations and suicide in post-COVID-19 syndrome.

Differences found between SUI and NON SUI suggest that presence of high risk of suicide might be associated with specific physical and psychopathological characteristics. These characteristics might have come together to increase the risk of suicide. The presence of higher percentages of subjects with psychiatric history and psychiatric history in relatives, as well as higher BPRS, HAM-D, HAM-A KMDRS, and BHS scores in SUI as compared to NON SUI confirm the well-known existent relationship between the presence of psychopathology and suicide [61]. In addition, greater percentages of physical complaints and comorbidities (as demonstrated by the higher percentages of subjects currently receiving medications) in SUI than NON SUI are in line with data from previous studies documenting the correlation between the presence of multiple major physical conditions and suicide risk [62]. High suicide rates have been also related to symptoms more frequently observed in SUI than NON SUI, such as pain conditions [63], autoimmune disorders [64,65], fever [66], respiratory system disorders [67], gastrointestinal disorders [68], chronic fatigue [69], vertigo [70], and anosmia [71].

The mechanism by which physical complaints and psychopathology might increase the risk of suicide are not known. The vast majority of studies assume that the presence of physical symptoms might cause distress and increase psychopathology, which, in turn, might raise the risk of suicide behaviors [72]. Nevertheless, an independent effect of physical distress and psychopathology symptoms has been reported [73]. A hyper-inflammatory state might represent a common mechanism linking post-COVID-19 syndrome, psychopathology, and suicide. Sustained inflammation has been indicated as the pathophysiological process of post-COVID-19 syndrome [74]. The principal mechanisms proposed are mast cell activation [75] and long-term autoimmunity [74]. More specifically, hyper-activation of CD8 T cells [76], increased blood levels of autoantibodies against phospholipids, interferons, neutrophils, connective tissues, cyclic citrullinated peptides, and cell nucleus [77,78], and increased levels of proinflammatory citokines such as interleukin (IL)-6, C-reactive-protein (CRP) [79], altered gut microbiota [80] have been ruled out. These alterations have been related to specific symptoms present in the post-COVID-19 syndrome and in those present in SUI, such as Sjogren syndrome [81], myalgia, fatigue [82], diarrhea [83], vertigo [70], anosmia [84], and cutaneous manifestations [85]. Hyperinflammation has been also hypothesized to underlie the pathophysiology of psychiatric syndromes, either in the context of COVID-19 or in subjects without physical complaints [86,87]. To this extent, IL6, TNFa, and CRP have been found to be abnormally elevated in sera samples of depressed suicide attempters [88], whereas elevated levels of peripheral blood mast cells have been related to increased suicide risk [89]. The aforementioned mechanisms have been hypothesized to exert a brain toxic effect in synergy with other pathologic processes, such as hyperactivation of brain macrophages. Toxicity might then be direct or mediated by glutamatergic/serotonergic imbalance [90].

Exploratory analyses revealed that a greater number of physical symptoms and greater levels of psychopathology differentiate HIGH SUI from LOW SUI. Differences in psychopathology were limited to BPRS and KMDRS scores. Therefore, mixed depression might be a factor increasing suicide risk. Mixed depression embeds the core features of depression, such as low mood, with excitatory symptoms, i.e., psychic and motor agitation, inner tension, insomnia, and racing thoughts [91]. Mixed depression has previously related to a high risk of suicide in non–COVID-19 samples [9] and has been thought to be driven by a brain inflammatory imbalance [92]. The presence of mixed depression in a context of post-COVID-19 has treatment implications. Research demonstrates the depressive episodes in the context of post-COVID-19 syndrome can be addressed with the use of antidepressants [93]. However, antidepressants have been proven to increase, rather than decrease, the severity of mixed depression, leading to higher rates of suicide [91]. Therefore, antidepressant treatments should be used cautiously while treating depressive symptoms in the context of post-COVID-19 syndrome.

Subjects with LOW SUI are more likely to undergo psychotherapy than those without. Psychotherapy has been associated with a reduced risk of suicide [94] and reduced psychopathology during and after SARS-CoV2 infection [95]. The biological mechanisms underlying psychotherapy are thought to involve anti-inflammatory mechanisms [96]. Therefore, while treating subjects with post-COVID-19 syndrome and with high suicide risk, psychotherapy might be considered a first-line strategy to treat psychopathology.

### Limitations

The definition of Post-COVID-19 syndrome provided by the present study is discordant with the one used in previous studies [4,97,98,99,100,101] or by agencies [102,103,104], which refers to the syndrome as the presence of specific symptoms lasting for definite periods of time. Nevertheless, to date, there is no universally accepted consensus on the definition of specific symptoms or their specific duration to define post-COVID-19 syndrome [33,34]. Therefore, in the absence of a consensus regarding such definition, the present study’s definition of post-COVID-19 syndrome should be considered as temporary.

The small sample size of SUI, HIGH SUI, and LOW SUI limits the generalizability of the results found. In particular, comparisons between HIGH SUI and LOW SUI should be considered as exploratory. Lager sample sizes regarding the risk of suicide are needed to corroborate the present findings. The cross-sectional nature of the present work impedes clear identification of the relationship among post-COVID-19 syndrome, psychopathology, and suicide. Therefore, the hypotheses made on the possible role of physical symptoms on psychopathology are only speculative. The present work limits its analysis to the physical and psychopathological symptoms/syndromes present during post-COVID-19 syndrome. Additional variables, such as income or presence/absence of isolation might better clarify the effect of post-COVID-19 syndrome on suicide. Furthermore, instrumental analyses, such as spirometry, the six-minute walking test, and a comprehensive cognitive battery might add a more precise measurement of impairment. Finally, the present study’s information on the psychiatric diagnosis and psychotropic drugs assumed is limited. Since such variables have proven to affect brain morphology and behavior [105,106,107,108], further studies are needed to evaluate the effect of psychiatric diagnosis and psychotropic drugs in subjects with post-COVID-19 syndrome.

Additionally, the small effect size regarding specific COVID-19 and post-COVID-19 syndrome’s symptoms should prompt caution in interpreting the differences found. Nevertheless, the medium effect size related to the between-group differences in the total amount of symptoms presented during and after COVID-19 gave reliability to the relationship between COVID-19-related physical burden and suicide risk. Larger sample sizes might be needed to better investigate the relationship between the presence of specific post-COVID-19 syndrome’s symptoms and suicide risk.

## 5. Conclusions

Subjects with suicide risk during post-COVID syndrome show higher levels of physical complaints and psychopathology. High risk of suicide is associated with multiple physical complaints, severe mixed depression, and the absence of psychotherapy. Interventions aimed to reduce physical distress and the use of psychotherapy might lower suicide risk. To explore suicidality in the post-COVID-19 syndrome, further studies are warranted.

## Figures and Tables

**Table 1 jpm-12-02019-t001:** Characteristics of subjects with SUI.

SUI	
(*n* = 41)	
High risk, *n* (%)	22 (53.6)
Presence of MDD, *n* (%)	36 (87.8)
Suicide attempt, *n* (%)	3 (7.3)
Suicide ideation, *n* (%)	21 (51.2)
Suicide plan, *n* (%)	12 (29.3)
Psychiatric history, *n* (%)	13 (31.7)

**Legend:** MDD, major depressive disorder; SUI, subjects with suicide risk.

**Table 2 jpm-12-02019-t002:** Demographic and clinical characteristics of subjects with SUI, NON SUI, LOW SUI, and HIGH SUI.

	NON SUI (*n* = 1547)	SUI (*n* = 41)			SUI vs. NON SUI			HIGH SUI vs. LOW SUI		
		OVERALL	HIGH SUI	LOW SUI	t	*p*	d/v	t	*p*	d/v
** Demographics **										
Age (y), mean ± SD	55.37 ± 14.35	51.71 ± 14.27	51.32 ± 12.75	52.16 ± 16.19	2.59	0.107	0.25	0.03	0.854	0.06
**Female, %**	46.7	65.9	68.2	63.2	5.89	**0.015**	0.06	0.11	0.735	0.05
Employed, %	66.2	61	59.1	63.2	0.48	0.486	0.17	0.07	0.790	0.04
Education (y), mean ± SD	13.36 ± 5.45	13.32 ± 3.78	12.95 ± 4.20	13.74 ± 3.28	0.002	0.963	0.01	0.43	0.515	0.21

**Legend:** Significant results are in **bold**. HIGH SUI, subjects with high suicide risk; LOW SUI, subjects with low suicide risk; NON SUI, subjects without suicide risk; SUI, subjects with suicide risk. d, Cohen’s d; SD, standard deviation; v, Cramer’s v.

**Table 3 jpm-12-02019-t003:** Clinical variables related to COVID-19 and post-COVID-19 syndrome in SUI, NON SUI, LOW SUI, and HIGH SUI.

Clinical										
	NON SUI (*n* = 1547)	SUI (*n* = 41)			SUI vs. NON SUI			HIGH SUI vs. LOW SUI		
		OVERALL	HIGH SUI	LOW SUI	t	*p*	d/v	t	*p*	d/v
Distance from COVID-19, mean ± SD	148.40 ± 105.53	155.51 ± 94.82	154.59 ± 89.76	156.58 ± 102.85	0.18	0.669	0.07			0.02
**Symptoms during COVID-19**										
Fever, %	82.9	90.2	81.8	100	1.52	0.217	0.03			0.31
Fatigue, %	81.5	92.7	95.5	89.5	3.35	0.067	0.05			0.12
Cough, %	60.2	65.9	77.3	52.6	0.52	0.469	0.02			0.26
**Diarrhea, %**	29.6	53.7	63.6	42.1	10.96	**0.001**	0.08	1.90	0.168	0.22
Headache, %	49.9	56.1	59.1	52.6	0.61	0.434	0.02			0.07
Anosmia, %	45.4	48.8	59.1	36.8	0.19	0.666	0.67			0.22
Dysgeusia, %	45.6	48.8	54.5	42.1	0.16	0.690	0.01			0.12
Red eyes, %	21.8	34.1	27.3	42.1	3.50	0.061	0.05			0.17
Low vision, %	19.7	24.4	31.8	15.8	0.55	0.459	0.02			0.19
Syncope, %	8.5	12.2	9.1	15.8	0.71	0.400	0.02			0.10
**Vertigo, %**	23.6	41.5	40.9	42.1	6.98	**0.008**	0.07	0.01	0.938	0.01
**Joint pain, %**	55.9	73.2	68.2	78.9	4.84	**0.028**	0.05	0.60	0.438	0.12
Skin lesions, %	11.4	14.6	18.2	10.5	0.39	0.527	0.02			0.11
Sjögren syndrome, %	23.4	34.1	31.8	36.8	2.55	0.110	0.04			0.05
**Raynaud’s phenomenon, %**	2.6	9.8	4.5	15.8	7.62	**0.006**	0.07	1.46	0.226	0.19
**Myalgia, %**	58.7	75.6	68.2	84.2	4.73	**0.030**	0.05	1.42	0.233	0.19
Dyspnea, %	67.1	73.2	77.3	68.4	0.67	0.413	0.02			0.10
**Chest pain, %**	37.1	61	59.1	63.2	9.69	**0.002**	0.08	0.07	0.790	0.04
Sore throat, %	27.6	39	36.4	42.1	2.59	0.107	0.04			0.06
Sputum, %	16.6	22	22.7	21.1	0.82	0.366	0.02			0.02
Rhinitis, %	25.7	31.7	22.7	42.1	0.76	0.383	0.02			0.21
Lack of appetite, %	42.9	51.2	36.4	68.4	1.12	0.290	0.03			0.04
**Number of symptoms during COVID-19, mean** **± SD**	8.38 ± 4.18	10.54 ± 4.49	10.45 ± 5.02	10.63 ± 3.93	10.60	**0.001**	0.49	0.02	0.902	0.04
Hospitalization, %	56.9	48.8	45.5	52.6	1.07	0.301	0.03			0.07
Length of hospitalization, mean ± SD	11.65 ± 21.67	11.37 ± 17.70	13.14 ± 21.68	9.32 ± 11.81	0.01	0.934	0.01			0.22
**Current symptoms (post COVID-19 symptoms)**										
**Fever, %**	1.4	9.8	13.6	5.3	17.23	**<0.001**	0.10	0.81	0.368	0.14
**Fatigue, %**	60.4	82.9	95.5	68.4	8.49	**0.004**	0.07	5.26	**0.022**	0.36
**Cough, %**	14	36.6	40.9	31.6	16.29	**<0.001**	0.10	0.38	0.536	0.10
**Diarrhea, %**	8.3	29.3	31.8	26.3	21.65	**<0.001**	0.12	0.15	0.699	0.06
**Headache, %**	20.7	46.3	59.1	31.6	15.55	**<0.001**	0.09	3.10	0.078	0.28
**Anosmia, %**	15.4	26.8	36.4	15.8	3.96	**0.047**	0.05	2.19	0.138	0.23
Dysgeusia, %	13.3	19.5	22.7	15.8	1.35	0.246	0.03			0.09
Red eyes, %	8.1	9.8	9.1	10.5	0.14	0.710	0.01			0.02
Low vision, %	19.8	29.3	50	5.3	2.21	0.137	0.04			0.49
Syncope, %	0.7	0	0	0	0.29	0.588	0.01			
**Vertigo, %**	13.2	24.4	31.8	15.8	4.30	**0.038**	0.05	1.42	0.233	0.19
**Joint pain, %**	33	56.1	59.1	52.6	9.52	**0.002**	0.08	0.17	0.678	0.07
**Skin lesions, %**	7.8	19.5	18.2	21.1	7.31	**0.007**	0.07	0.05	0.817	0.04
**Sjögren syndrome, %**	12.2	24.4	40.9	5.3	5.48	**0.019**	0.06	7.02	**0.008**	0.41
Raynaud phenomenon, %	1.4	2.4	0	5.3	0.29	0.591	0.01			0.17
**Myalgia, %**	31.4	51.2	72.7	26.3	7.21	**0.007**	0.07	8.79	**0.003**	0.46
Dyspnea, %	61.2	68.3	81.8	52.6	0.86	0.354	0.02			0.31
Chest pain, %	20.7	29.3	31.8	26.3	1.75	0.186	0.03			0.06
Sorethroat, %	5.6	9.8	13.6	5.3	1.32	0.251	0.03			0.14
Sputum, %	6.6	37.3	9.1	5.3	0.03	0.854	0.01			0.07
Rhinitis, %	8.3	14.6	13.6	15.8	2.09	0.148	0.04			0.03
Lack of appetite, %	6.5	7.3	9.1	5.3	0.04	0.840	0.01			0.07
Deficit attention/concentration %	15	13.2	5	22.2	0.19	0.748	0.01			0.25
**Number of symptoms, mean** **± SD**	3.70 ± 3.12	6.05 ± 3.83	7.41 ± 3.91	4.47 ± 3.13	22.33	**<0.001**	0.67	6.88	**0.012**	0.83
**Current medications, %**	67.1	87.8	86.4	89.5	7.82	**0.005**	0.07	0.09	0.762	0.05
Current cardiac medications, %	7.5	2.6	0	5.9	1.29	0.256	0.03			0.18
**Current pneumological medications, %**	1	5.3	9.5	0	6.35	**0.012**	0.07	1.70	0.191	0.21
**Current endocrinological medications, %**	5.4	15.8	19	11.8	7.41	**0.006**	0.07	0.37	0.540	0.10
Current rheumatological medications, %	0.8	0	0	0	0.29	0.588	0.01			
Other current medications, %	9.5	10.5	9.5	11.8	0.05	0.824	0.01			0.04
Current polytherapy, %	40.5	52.6	47.6	58.8	2.27	0.132	0.04			0.11
Number of current medications, mean ± SD	2.27 ± 2.68	3.10 ± 3.43	4.09 ± 3.87	1.95 ± 2.46	3.79	0.052	0.27	4.32	0.044	0.66

**Legend:** Significant results are in **bold**. HIGH SUI, subjects with high suicide risk; LOW SUI, subjects with low suicide risk; NON SUI, subjects without suicide risk; SUI, subjects with suicide risk. d, Cohen’s d; SD, standard deviation; v, Cramer’s v.

**Table 4 jpm-12-02019-t004:** Psychopathological characteristics of subjects with SUI, NON SUI, LOW SUI, and HIGH SUI.

	NON SUI (*n* = 1547)	SUI (*n* = 41)			SUI vs. NON SUI			HIGH SUI vs. LOW SUI		
		OVERALL	HIGH SUI	LOW SUI	t	*p*	d/v	t	*p*	d/v
**Current psychotropic drugs, %**	8.5	17.1	13.6	21.1	3.73	**0.054**	0.05			0.10
**Psychiatric history, %**	8.9	31.7	22.7	42.1	24.38	**<0.001**	0.12	1.77	0.184	0.21
**Psychiatric history in relatives, %**	8.9	22	9.1	36.8	8.20	**0.004**	0.07	4.58	**0.032**	0.33
**Previous psychopharmacotherapy, %**	5.3	19.5	18.2	21.1	15.09	**<0.001**	0.09	0.05	0.817	0.04
**Previous psychotherapy, %**	6.4	22	9.1	36.8	15.24	**<0.001**	0.09	4.58	**0.032**	0.33
Previous use of substances, %	17.1	14.6	4.5	26.3	0.17	0.683	0.01			0.31
**BPRS, mean** **± SD**	26.34 ± 4.22	33.54 ± 8.20	36 ± 9.98	30.68 ± 4.16	108.45	**<0.001**	1.10	4.67	**0.037**	0.69
**HAM-A, mean** **± SD**	4.94 ± 5.42	13.24 ± 8.08	13.27 ± 7.56	13.21 ± 8.86	91.09	**<0.001**	1.20	0.00	0.981	0.01
**HAM-D, mean** **± SD**	3.77 ± 4.19	12.24 ± 6.35	13.14 ± 6.39	11.21 ± 6.31	157.71	**<0.001**	1.57	0.94	0.339	0.30
**KMDRS, mean** **± SD**	5.33 ± 2.45	9.88 ± 4.32	11.18 ± 4.55	8.37 ± 3.58	131.23	**<0.001**	1.29	4.73	**0.036**	0.68
BHS, mean ± SD	9.79 ± 2.42	10.67 ± 2.51	10.73 ± 2.79	10.58 ± 2.23	3.44	0.064	0.36			0.06
**BHS-future, mean** **± SD**	2.20 ± 1.89	3.96 ± 1.23	2.29 ± 1.89	2.09 ± 1.97	48.84	**<0.001**	1.10	0.06	0.805	0.10
**BHS-motivation, mean** **± SD**	2.17 ± 1.36	3.88 ± 1.71	3.93 ± 1.82	3.82 ± 1.66	37.88	**<0.001**	1.26	0.02	0.877	0.06
BHS-expectation, mean ± SD	3.42 ± 7.59	4.08 ± 6.38	5.07 ± 8.41	2.82 ± 1.66	0.18	0.668	0.09			0.37

**Legend:** Significant results are in **bold**. BHS, Beck hopelessness scale; BPRS, brief-psychiatric rating scale; HAM-A, Hamilton anxiety rating scale; HAM-D, Hamilton rating scale for depression; KMDRS, Koukopoulos mixed depression rating scale; HIGH SUI, subjects with high suicide risk; LOW SUI, subjects with low suicide risk; NON SUI, subjects without suicide risk; SUI, subjects with suicide risk. d, Cohen’s d; SD, standard deviation; v, Cramer’s v.

## Data Availability

A file with anonymized participant data will be provided to those making a reasonable request to the corresponding author.

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
