# Peer review of "Suicide Risk in Post-COVID-19 Syndrome"

_jpm, 2022, doi:10.3390/jpm12122019_

Round 1
Reviewer 1 Report
The manuscript titled “Suicide risk in the post-COVID-19 syndrome” aimed to clarify the relationship between post-COVID-19 syndrome and suicide risk. The analyses revealed high risk of suicide was associated with multiple physical complaints, severe mixed depression, and absence of psychotherapy. This study may be helpful for us to understand risk of suicide in subjects with post-COVID-19 syndrome. However, there are several concerns in the manuscript.
Comments
#1. In the Introduction section, the authors made little mention of the characteristics of people who are vulnerable to suicide risk during the COVID-19 pandemic. There are some articles on this topic. The authors should provide a more detailed description on such individuals.
#2. In the Methods section, who did the psychopathological evaluation? A trained psychiatrist or psychologist? Explain it.
#3. The authors described in the Results section as follows: “As regards sociodemographic characteristics, SUI showed higher rates of women than NON SUI.” However, there is no description of this result in the discussion. This is an important result and should be discussed. The following article may be useful:
Takubo Y et al. Changes in thoughts of self-harm among postpartum mothers during the prolonged COVID-19 pandemic in Japan. Psychiatry Clin Neurosci. 2022; 76(10): 528-529.
#4. The authors described in the Discussion section as follows: “The larger part of this study patients’ enrollment ranged from April 2020 to April 2021, i.e., the first year of COVID-19 pandemic.” How many subjects enrolled during this period? The authors should add this information in the results.
#5. The authors described in the Discussion section as follows: “Authors should discuss the results and how they can be interpreted from the perspective of previous studies and of the working hypotheses. The findings and their implications should be discussed in the broadest context possible. Future research directions may also be highlighted.” This is a template for authors and should be removed.
#6. Did the researchers provide care for those who were determined to be at high risk of suicide? Clarify it.
Author Response
Response to Reviewer 1 Comments
Point 1: In the Introduction section, the authors made little mention of the characteristics of people who are vulnerable to suicide risk during the COVID-19 pandemic. There are some articles on this topic. The authors should provide a more detailed description on such individuals.
Response 1: We thank Reviewer 1 for such suggestion. We provided a more detailed description of results of other studies.
“Five studies [14-19] focused on possible risk factors for suicidality. Social distancing due to quarantine, unemployment, safety concerns, being unmarried, previous psychopathology were found to be mostly associated to suicide.
Others [20-24] mainly focused on recommendations for suicide prevention. Strategies proposed were: I) developing a systematic suicide screening test; II) facilitating communication and increasing access to the interventions for the people at risk; III) promoting mental health programs; IV) implementing measures to mitigate the economic breakdown; V) regulating media reporting.”
Point 2: In the Methods section, who did the psychopathological evaluation? A trained psychiatrist or psychologist? Explain it.
Response 2: We thank Reviewer 1 for suggesting this clarification. The assessment was made by a trained psychiatrist. We emphasized the study evaluation protocol in the appropriate section, as follows:
“The study was conducted by all the authors in the context of Gemelli Against COVID-19 Post-Acute Care Study Group. The group was formed with the aim of studying the post-acute sequelae of SARS-Cov2 infection.”
“Patients underwent a comprehensive assessment including collection of detailed medical history and a thorough physical examination. Additionally, they received internal medicine, geriatric, ophthalmological, otolaryngologic, pneumological, psychiatric, cardiological, immunological, and rheumatological evaluations. Specifically, anamnesis regarding demographical characteristics, severity and course of COVID-19 infection, together with a general medical examination was performed by an internal medicine specialist. geriatric, ophthalmological, otolaryngologic, pneumological, psychiatric, cardiological, immunological, and rheumatological evaluations were provided by specialists of the Gemelli Against COVID-19 Post-Acute Care Service.
As regards psychiatric evaluation, a senior psychiatrist (AS) performed diagnostic assessments using rating scales assessing depression, suicidal risk, severity of general psychopathology, severity of depressive symptoms; severity of anxiety symptoms, severity of mixed depression, hopelessness. Then, a minimum of two trained diagnosticians (postdoctoral researchers) were additionally involved in the diagnostic process. Specifically, diagnostic consensus among the treating psychiatrist, the senior psychiatrist and the trained diagnosticians was needed to include the subject in the study. In the event of diagnostic disagreement between one or more subject belonging to the diagnostic team, the participant was excluded from the study. Prior to the start of the study, a specific training on rating scales administration was provided. Then, psychometricians performed practice assessments, and received performance reviews and feedback on test administration. Their training process continued until an inter-rater reliability of at least 0.80 (κ coefficient) was reached”.
Point 3: The authors described in the Results section as follows: “As regards sociodemographic characteristics, SUI showed higher rates of women than NON SUI.” However, there is no description of this result in the discussion. This is an important result and should be discussed. The following article may be useful: Takubo Y et al. Changes in thoughts of self-harm among postpartum mothers during the prolonged COVID-19 pandemic in Japan. Psychiatry Clin Neurosci. 2022; 76(10): 528-529.
Response 3: We thank Reviewer 1 for this suggestion. We added a dedicated paragraph in the Discussion section, as follows:
“The higher rates of women we found within SUI than NON SUI might seem in contrast with the well-known finding of a higher suicide rate in men as compared to women [58]. Such discrepancy has been described before as the “gender paradox in suicide”. The paradox resides in the fact that females show more suicide thoughts and attempt suicide more frequently than males do, whereas males have higher rates of suicide [59]. As shown by a Japanese study of Takubo and collegues on suicidal ideation among portpartum mothers during COVID-19 [60], this phenomenon has been exacerbated by the pandemic, maybe due to the decrease in female employment and increase in domestic violence [61]”.
Point 4: The authors described in the Discussion section as follows: “The larger part of this study patients’ enrollment ranged from April 2020 to April 2021, i.e., the first year of COVID-19 pandemic.” How many subjects enrolled during this period? The authors should add this information in the results.
Response 4: We thank Reviewer 1 for this comment. We specified the exact number of subjects enrolled in the first year. The sentence provided in the text is shown below:
“The larger part of this study patients’ enrollment, ranged from April 2020 to April 2021, i.e., the first year of COVID-19 pandemic. Specifically, 1261 subjects (81.5% of the whole sample) were assessed in this period”.
Point 5: The authors described in the Discussion section as follows: “Authors should discuss the results and how they can be interpreted from the perspective of previous studies and of the working hypotheses. The findings and their implications should be discussed in the broadest context possible. Future research directions may also be highlighted.” This is a template for authors and should be removed.
Response 5: We apologize with Reviewer 1 for such oversight. We deleted this template.
Point 6: Did the researchers provide care for those who were determined to be at high risk of suicide?Clarify it.
Response 6: We thank Reviewer 1 for requesting clarification on this aspect. We provided care for those with suicide risk. A brief description of the care management of these patients is provided below and will provided in the text as well.
“In case of any suicide risk, we provided tailored care in accordance with the European Psychiatric Association Guidelines for Suicide Prevention [46]. Specifically, we immediately evaluated the patient and implemented a treatment plan focused on reducing acute psychiatric symptoms such as anxiety, insomnia, depression, and eventual psychotic symptoms. In severe cases, or when the patient was not surrounded by a social network, we opted for hospitalization in a psychiatric ward.
When we could send the patient home, we involved family members in the treatment plan, asking them to do not leave the patient alone. Then, we increased surveillance by providing more frequent visits and telephone contacts to monitor progress and possible side effects of medication. Adjustment of the aforementioned guidelines was made according to COVID-19 restrictions [47]”.
Reviewer 2 Report
The aim of the present study was to evaluate the risk of suicide in subjects with post-COVID-19 syndrome. For this purpose, 1588 patients with a previous history of COVID syndrome were included and underwent psychopathological evaluation along with several psychometric tests for anxiety, depression, general psychopathology and specifically for suicide. Detailed evaluations and large sample size are strengths of the study. The study revealed that greater levels of physical complaints and psychopathology during post-COVID-syndrome might increase the risk to commit suicide. After several improvements, the study might be beneficial to readers and community.
1. Some points should be clarified in the study title and objectives. Was this study done with patients with post COVID syndrome or was it done with former patients who had COVID and had post COVID symptoms. My impression is that the study was done with the second group. The inclusion criteria did not include that they had post COVID syndrome. In this case, it seems more accurate to say patients with post COVID symptoms. This should be corrected both in the title and in the content.
2. If the patients were diagnosed as having post COVID syndrome, it should be explained in detail according to which criteria and based on which symptoms they had.
3. Explain how the sample size was found, how many people were approached and how many people were excluded and why. Power analysis findings for the sample size should be included.
4. The implementation of the study should be explained in detail. Who conducted the study, who conducted the interviews, how long they lasted, what were their competencies.
5. The characteristics of the scales used should be given in more detail. Why they were preferred should be emphasized. Psychometric properties of both the original and adapted scales should be included.
6. The discussion is generally adequate, especially the finding about mixed depression is a valuable element in terms of literature and should be emphasized in further studies.
7. There are some grammatical and spelling errors in the manuscript, which should be corrected.
Author Response
Response to Reviewer 2 Comments
Point 1: Some points should be clarified in the study title and objectives. Was this study done with patients with post COVID syndrome or was it done with former patients who had COVID and had post COVID symptoms. My impression is that the study was done with the second group. The inclusion criteria did not include that they had post COVID syndrome. In this case, it seems more accurate to say patients with post COVID symptoms. This should be corrected both in the title and in the content.
Response 1: We Thank Reviewer 2 for raising this issue. We are aware of the existing debate on the possible criteria used to establish the presence of a post-COVID-19 syndrome. To date, there is still not a universally accepted consensus in the definition of post-COVID-19 syndrome (Munblit et al., 2022). Various agencies have generated their own terms and definitions, including WHO (https://www.thelancet.com/journals/ laninf/article/PIIS1473-3099(21) 00703-9/fulltext), the UK National Institute for Health and Care Excellence (https://www.nice.org.uk/guidance/ng188/resources/covid19-rapid-guideline-managing-the-longterm-effects-of-covid19-pdf-51035515742, https://www.nice.org.uk/guidance/ ng188/resources/covid19-rapid-guideline-managing-the-longterm-effects-of-covid19-pdf-51035515742), and the US Centes for Disease Control and Prevention (https://www.cdc.gov/coronavirus/2019-ncov/long-term-effects/index.html). As regards research, the definition is still widely used by researchers as a very broad term covering persistent signs and symptoms that continue or develop after acute SARS-CoV-2 infection. Symptoms presentation and duration after COVID-19 is still very heterogeneous (Callard et al, 2021; Altmann et al, 2021; Boyton et 2021; Baig et al, 2020; Venkatesan et al., 202; Brodin et al, 2021; Davido B, et al. 2020; Datta et al, 2020; Mendelson M, Nel J, Blumberg L, et al. 2020; Sivan et al, 2020; Nalbandian A, Sehgal K, Gupta A, et al. 2021; Fernandez-de-Las-Penas C, Palacios-Cena D, Gomez-Mayordomo V, et al. 2021; Greenhalgh T, Knight M, A’Court C, et al, 2020; Shah W, Hillman T, Playford ED, et al, 2021).
In consideration of such heterogeneity, we decided to use the definition of post-COVID-19 syndrome to indicate all conditions characterized by the persistence of COVID-19 symptoms after negativization. Accordingly, we prefer to maintain such definition throughout the text.
However, Reviewer 2’s question highlighted an unsolved issue in the scientific community. Furthermore, we agree with Reviewer 2 in the fact that confusion might raise while defining Post-COVID-19 syndrome. Therefore, we provided a specific paragraph regarding such issue in the Limitation Section, as follows:
“Definition of Post-COVID-19 syndrome provided by the present study is discordant with the one of previous studies [100-105] or agencies [106-108], which refer to such syndrome as presence of specific symptoms lasting for definite periods of time. Nevertheless, to date, there is still not a universally accepted consensus in the definition of specific symptoms or their specific duration to define post-COVID-19 syndrome [109,110]. Therefore, in wait of a consensus regarding such definition, the present study’s definition of post-COVID-19 syndrome should be considered as temporary”.
Point 2: If the patients were diagnosed as having post COVID syndrome, it should be explained in detail according to which criteria and based on which symptoms they had.
Response 2: We thank Reviewer 2 for this suggestion. We provided a detailed explanation below.
“In accordance with most published research [27-32] and due to the lack of homogeneity in the proposed criteria for the definition of the post-COVID-19 syndrome [33,34] we defined as Post-COVID-19 syndrome any condition characterized by the presence of one or more symptoms that persisted after negativization, regardless of symptom’s duration or severity”.
Point 3: Explain how the sample size was found, how many people were approached and how many people were excluded and why. Power analysis findings for the sample size should be included.
Response 3: We thank Reviewer 2 for highlighting this aspect. We provided a more detailed description of the sample selection process, as follows:
“Two thousand five hundred fifty-nine subjects accessed the Post-Acute Care Service of the Fondazione Policinico Universitario Agostino Gemelli IRCCS. Two hundred nine patients were excluded due to unsuitable age range. Six subjects were excluded because of dementia. Seven-hundred fifty-six subjects were unable to complete the whole evaluation and were excluded from the study. Therefore, the final sample consisted of 1588 subjects”.
We also provided a description of power analyses performed in the results and limitation section. The added paragraphs are shown below:
“Effect sizes regarding differences in psychopathology were strong, whereas those regarding number of symptoms during and after COVID-19 are medium. On the other hand, effect sizes related to differences in demographical and clinical variables, and specific COVID-19 and post-COVID-19 syndrome’s symptoms were small”.
“Additionally, the small effect size regarding specific COVID-19 and post-COVID-19 syndrome’s symptoms should prompt caution in interpreting differences found. Nevertheless, the medium effect size related to the between-group differences in the total amount of symptoms presented during and after COVID-19 gave reliability to the relationship between COVID-19 related physical burden and suicide risk. Larger sample sizes might be needed to better investigate the relationship between presence of specific Post-COVID-19-syndrome’s symptoms and suicide risk”.
Point 4: The implementation of the study should be explained in detail. Who conducted the study, who conducted the interviews, how long they lasted, what were their competencies.
Response 4: We thank Reviewer 2 for raising this issue. A more detailed description of the study is provided below.
“The study was conducted by all the authors in the context of Gemelli Against COVID-19 Post-Acute Care Study Group. The group was formed with the aim of studying the post-acute sequelae of SARS-Cov2 infection”.
“Patients underwent a comprehensive assessment including collection of detailed medical history and a thorough physical examination. Additionally, they received internal medicine, geriatric, ophthalmological, otolaryngologic, pneumological, psychiatric, cardiological, immunological, and rheumatological evaluations. Specifically, anamnesis regarding demographical characteristics, severity and course of COVID-19 infection, together with a general medical examination was performed by an internal medicine specialist. geriatric, ophthalmological, otolaryngologic, pneumological, psychiatric, cardiological, immunological, and rheumatological evaluations were provided by specialists of the Gemelli Against COVID-19 Post-Acute Care Service.
As regards psychiatric evaluation, a senior psychiatrist (AS) performed diagnostic assessments using rating scales assessing depression, suicidal risk, severity of general psychopathology, severity of depressive symptoms; severity of anxiety symptoms, severity of mixed depression, hopelessness. Then, a minimum of two trained diagnosticians (postdoctoral researchers) were additionally involved in the diagnostic process. Specifically, diagnostic consensus among the treating psychiatrist, the senior psychiatrist and the trained diagnosticians was needed to include the subject in the study. In the event of diagnostic disagreement between one or more subject belonging to the diagnostic team, the participant was excluded from the study. Prior to the start of the study, a specific training on rating scales administration was provided. Then, psychometricians performed practice assessments, and received performance reviews and feedback on test administration. Their training process continued until an inter-rater reliability of at least 0.80 (κ coefficient) was reached”.
Point 5: The characteristics of the scales used should be given in more detail. Why they were preferred should be emphasized. Psychometric properties of both the original and adapted scales should be included.
Response 5: We thank Reviewer 2 for highlighting this aspect. We gave a more detailed description of psychometric properties of scales used and gave a more detailed description of the rationale behind the scales’ selection. The added paragraphs were the following:
“These domains were assessed with rating scales largely used in common clinical practice as well as in research. The aforementioned psychopathologic assessment was chosen because of the close relationship to suicide risk and attempts [8-10]. Detailed description of psychiatric rating scales used was provided below.
“MINI-suicidal assesses suicidal risk. We used the short form because of its suitability for screening in an acute setting”.
“The Brief Psychiatric Rating Scale (BPRS) [39]. BPRS was developed as a measurement general psychopathology. It assesses a big variety of psychiatric symptoms, such as anxiety, depression, and psychosis”.
“The Hamilton Rating Scale for Depression (HAM-D)[40]. HAM-D is used to assess severity of depressive symptoms. We used the original version containing 17 items. Each item is related to depressive symptoms experienced over the past week. Each item is scored between 0-4 or 0-2, depending on the item”.
“The Hamilton Anxiety Rating Scale (HAM-A) [41]. HAM-A is a questionnaire used to measure the severity of anxiety symptoms. The scale consists of 14 items and measures both psychic anxiety (worries, irritability, feelings of tension, etc.) and somatic anxiety (physical complaints related to anxiety)”.
Point 6: The discussion is generally adequate, especially the finding about mixed depression is a valuable element in terms of literature and should be emphasized in further studies.
Response 6: We thank Reviewer 2 for this comment.
Point 7: There are some grammatical and spelling errors in the manuscript, which should be corrected.
Response 7: We apologize for the typos made. We corrected all the grammatical and spelling errors throughout the manuscript.
Round 2
Reviewer 1 Report
The current manuscript has been appropriately revised based on reviewer comments and is suitable for publication.